# Structure learning of antiferromagnetic Ising models

**Guy Bresler[1]**    **David Gamarnik[2]**    **Devavrat Shah[1]**
Laboratory for Information and Decision Systems
Department of EECS[1] and Sloan School of Management[2]
Massachusetts Institute of Technology
{gbresler,gamarnik,devavrat}@mit.edu

## Abstract

In this paper we investigate the computational complexity of learning the graph structure underlying a discrete undirected graphical model from i.i.d. samples. Our first result is an unconditional computational lower bound of $\Omega(p^{d/2})$ for learning general graphical models on $p$ nodes of maximum degree $d$, for the class of so-called statistical algorithms recently introduced by Feldman et al. [1]. The construction is related to the notoriously difficult learning parities with noise problem in computational learning theory. Our lower bound suggests that the $\widetilde{O}(p^{d+2})$ runtime required by Bresler, Mossel, and Sly's [2] exhaustive-search algorithm cannot be significantly improved without restricting the class of models.

Aside from structural assumptions on the graph such as it being a tree, hypertree, tree-like, etc., many recent papers on structure learning assume that the model has the correlation decay property. Indeed, focusing on ferromagnetic Ising models, Bento and Montanari [3] showed that all known low-complexity algorithms fail to learn simple graphs when the interaction strength exceeds a number related to the correlation decay threshold. Our second set of results gives a class of repelling (antiferromagnetic) models that have the *opposite* behavior: very strong interaction allows efficient learning in time $\widetilde{O}(p^2)$. We provide an algorithm whose performance interpolates between $\widetilde{O}(p^2)$ and $\widetilde{O}(p^{d+2})$ depending on the strength of the repulsion.

## 1   Introduction

Graphical models have had tremendous impact in a variety of application domains. For unstructured high-dimensional distributions, such as in social networks, biology, and finance, an important first step is to determine which graphical model to use. In this paper we focus on the problem of structure learning: Given access to $n$ independent and identically distributed samples $\sigma^{(1)}, \ldots \sigma^{(n)}$ from an undirected graphical model representing a discrete random vector $\sigma = (\sigma_1, \ldots, \sigma_p)$, the goal is to find the graph $G$ underlying the model. Two basic questions are 1) How many samples are required? and 2) What is the computational complexity?

In this paper we are mostly interested in the computational complexity of structure learning. We first consider the problem of learning a general discrete undirected graphical model of bounded degree.

## 1.1 Learning general graphical models

Several algorithms based on exhaustively searching over possible node neighborhoods have appeared in the last decade [4, 2, 5]. Abbeel, Koller, and Ng [4] gave algorithms for learning general graphical models close to the true distribution in Kullback-Leibler distance. Bresler, Mossel, and Sly [2] presented algorithms guaranteed to learn the true underlying graph. The algorithms in both [4] and [2] perform a search over candidate neighborhoods, and for a graph of maximum degree $d$, the computational complexity for recovering a graph on $p$ nodes scales as $\widetilde{O}(p^{d+2})$ (where the $\widetilde{O}$ notation hides logarithmic factors).

While the algorithms in [2] are guaranteed to reconstruct general models under basic nondegeneracy conditions using an optimal number of samples $n = O(d \log p)$ (sample complexity lower bounds were proved by Santhanam and Wainwright [6] as well as [2]), the exponent $d$ in the $\widetilde{O}(p^{d+2})$ run-time is impractically high even for constant but large graph degrees. This has motivated a great deal of work on structure learning for special classes of graphical models. But before giving up on general models, we ask the following question:

**Question 1:** Is it possible to learn the structure of general graphical models on $p$ nodes with maximum degree $d$ using substantially less computation than $p^d$?

Our first result suggests that the answer to Question 1 is negative. We show an unconditional computational lower bound of $p^{\frac{d}{2}}$ for the class of *statistical algorithms* introduced by Feldman et al. [1]. This class of algorithms was introduced in order to understand the apparent difficulty of the Planted Clique problem, and is based on Kearns' statistical query model [7]. Kearns showed in his landmark paper that statistical query algorithms require exponential computation to learn parity functions subject to classification noise, and our hardness construction is related to this problem. Most known algorithmic approaches (including Markov chain Monte Carlo, semidefinite programming, and many others) can be implemented as statistical algorithms, so the lower bound is fairly convincing.

We give background and prove the following theorem in Section 4.

**Theorem 1.1.** *Statistical algorithms require at least $\Omega(p^{\frac{d}{2}})$ computation steps in order to learn the structure of a general graphical models of degree $d$.*

If complexity $p^d$ is to be considered intractable, what shall we consider as tractable? Writing algorithm complexity in the form $c(d)p^{f(d)}$, for high-dimensional (large $p$) problems the exponent $f(d)$ is of primary importance, and we will think of tractable algorithms as having an $f(d)$ that is bounded by a constant independent of $d$. The factor $c(d)$ is also important, and we will use it to compare algorithms with the same exponent $f(d)$.

In light of Theorem 1.1, reducing computation below $p^{\Omega(d)}$ requires restricting the class of models. One can either restrict the graph structure or the nature of the interactions between variables. The seminal paper of Chow and Liu [8] makes a model restriction of the first type, assuming that the graph is a tree; generalizations include to polytrees [9], hypertrees [10], and others. Among the many possible assumptions of the second type, the correlation decay property is distinguished: to the best of our knowledge all existing low-complexity algorithms require the correlation decay property [3].

## 1.2 Correlation decay property

Informally, a graphical model is said to have the correlation decay property (CDP) if any two variables $\sigma_s$ and $\sigma_t$ are asymptotically independent as the graph distance between $s$ and $t$ increases. Exponential decay of correlations holds when the distance from independence decreases exponentially fast in graph distance, and we will mean this stronger form when referring to correlation decay. Correlation decay is known to hold for a number of pairwise graphical models in the so-called high-temperature regime, including Ising, hard-core lattice gas, Potts (multinomial) model, and others (see, e.g., [11, 12, 13, 14, 15, 16]).

Bresler, Mossel, and Sly [2] observed that it is possible to efficiently learn models with (exponential) decay of correlations, under the additional assumption that neighboring variables have correlation bounded away from zero (as is true, e.g., for the ferromagnetic Ising model in the high temperature regime). The algorithm they proposed for this setting pruned the candidate set of neighbors for each node to roughly size $O(d)$ by retaining only those variables with sufficiently high correlations, and then within this set performed the exhaustive search over neighborhoods mentioned before, resulting in a computational cost of $d^{O(d)}\widetilde{O}(p^2)$. The greedy algorithms of Netrapali et al. [17] and Ray et al. [18] also require the correlation decay property and perform a similar pruning step by retaining only nodes with high pairwise correlation; they then use a different method to select the true neighborhood.

A number of papers consider the problem of reconstructing Ising models on graphs with few short cycles, beginning with Anandkumar et al. [19]. Their results apply to the case of Ising models on sparsely connected graphs such as the Erdös-Renyi random graph $\mathcal{G}(p, \frac{d}{p})$. They additionally require the interaction parameters to be either generic or ferromagnetic. Ferromagnetic models have the benefit that neighbors always have a non-negligible correlation because the dependencies cannot cancel, but in either case the results still require the CDP to hold. Wu et al. [20] remove the assumption of generic parameters in [19], but again require the CDP.

Other algorithms for structure learning are based on convex optimization, such as Ravikumar et al.'s [21] approach using regularized node-wise logistic regression. While this algorithm does not explicitly require the CDP, Bento and Montanari [3] found that the logistic regression algorithm of [21] provably fails to learn certain ferromagnetic Ising model on simple graphs without correlation decay. Other convex optimization-based algorithms such as [22, 23, 24] require similar incoherence or restricted isometry-type conditions that are difficult to verify, but likely also require correlation decay. Since all known algorithms for structure learning require the CDP, we ask the following question (paraphrasing Bento and Montanari):

**Question 2:** Is low-complexity structure learning possible for models which do not exhibit the CDP, on general bounded degree graphs?

Our second main result answers this question affirmatively by showing that a broad class of repelling models on general graphs can be learned using simple algorithms, even when the underlying model does not exhibit the CDP.

## 1.3 Repelling models

The antiferromagnetic Ising model has a negative interaction parameter, whereby neighboring nodes prefer to be in opposite states. Other popular antiferromagnetic models include the Potts or coloring model, and the hard-core model.

Antiferromagnetic models have the interesting property that correlations between neighbors can be zero due to cancellations. Thus algorithms based on pruning neighborhoods using pairwise correlations, such as the algorithm in [2] for models with correlation decay, does not work for anti-ferromagnetic models. To our knowledge there are no previous results that improve on the $p^d$ computational complexity for structure learning of antiferromagnetic models on general graphs of maximum degree $d$.

Our first learning algorithm, described in Section 2, is for the hard-core model.

**Theorem 1.2** (Informal)**.** *It is possible to learn strongly repelling models, such as the hard-core model, with run-time $\widetilde{O}(p^2)$.*

We extend this result to weakly repelling models (equivalent to the antiferromagnetic Ising model parameterized in a nonstandard way, see Section 3). Here $\beta$ is a repelling strength and $h$ is an external field.

**Theorem 1.3** (Informal)**.** *Suppose $\beta \geq (d - \alpha)(h + \ln 2)$ for an integer $0 \leq \alpha < d$. Then it is possible to learn a repelling model with interaction $\beta$, with run-time $\widetilde{O}(p^{2+\alpha})$.*

The computational complexity of the algorithm interpolates between $\widetilde{O}(p^2)$, achievable for strongly repelling models, and $\widetilde{O}(p^{d+2})$, achievable for general models using exhaustive search. The complexity depends on the repelling strength of the model, rather than structural assumptions on the graph as in [19, 20].

We remark that the strongly repelling models exhibit long-range correlations, yet the algorithmic task of graph structure learning is possible using a local procedure.

The focus of this paper is on structure learning, but the problem of parameter estimation is equally important. It turns out that the structure learning problem is strictly more challenging for the models we consider: once the graph is known, it is not difficult to estimate the parameters with low computational complexity (see, e.g., [4]).

## 2 Learning the graph of a hard-core model

We warm up by considering the hard-core model. The analysis in this section is straightforward, but serves as an example to highlight the fact that correlation decay is not a necessary condition for structure learning.

Given a graph $G = (V, E)$ on $|V| = p$ nodes, denote by $\mathcal{I}(G) \subseteq \{0, 1\}^p$ the set of independent set indicator vectors $\sigma$, for which at least one of $\sigma_i$ or $\sigma_j$ is zero for each edge $\{i, j\} \in E(G)$. The hardcore model with fugacity $\lambda > 0$ assigns nonzero probability only to vectors in $\mathcal{I}(G)$, with

$$\mathsf{P}(\sigma) = \frac{\lambda^{|\sigma|}}{Z}, \quad \sigma \in \mathcal{I}(G). \tag{2.1}$$

Here $|\sigma|$ is the number of entries of $\sigma$ equal to one and $Z = \sum_{\sigma \in \mathcal{I}(G)} \lambda^{|\sigma|}$ is the normalizing constant called the partition function. If $\lambda > 1$ then more mass is assigned to larger independent sets. (We use indicator vectors to define the model in order to be consistent with the antiferromagnetic Ising model in the next section.)

Our goal is to learn the graph $G = (V, E)$ underlying the model (2.1) given access to independent samples $\sigma^{(1)}, \ldots, \sigma^{(n)}$. The following simple algorithm reconstructs $G$ efficiently.

---

**Algorithm 1** SIMPLEHC$(\sigma^{(1)}, \ldots, \sigma^{(n)})$

---

1: FOR each $i, j, k$:
2:   IF $\sigma_i^{(k)} = \sigma_j^{(k)} = 1$, THEN $S = S \cup \{i, j\}$
3: OUTPUT $\hat{E} = S^c$

---

The idea behind the algorithm is very simple. If $\{i, j\}$ belongs to the edge set $E(G)$, then for every sample $\sigma^{(k)}$ either $\sigma_i^{(k)} = 0$ or $\sigma_j^{(k)} = 0$ (or both). Thus for every $i, j$ and $k$ such that $\sigma_i^{(k)} = \sigma_j^{(k)} = 1$ we can safely declare $\{i, j\}$ *not* to be an edge. To show correctness of the algorithm it is therefore sufficient to argue that for every non-edge $\{i, j\}$ there is a high likelihood that such an independent set $\sigma^{(k)}$ will be sampled.

Before doing this, we observe that SIMPLEHC actually computes the maximum-likelihood estimate for the graph $G$. To see this, note that an edge $e = \{i, j\}$ for which $\sigma_i^{(k)} = \sigma_j^{(k)} = 1$ for some $k$ cannot be in $\hat{G}$, since $\mathsf{P}(\sigma^{(k)}|\hat{G}+e) = 0$ for any $\hat{G}$. Thus the ML estimate contains a subset of those edges $e$ which have not been ruled out by $\sigma^{(1)}, \ldots, \sigma^{(n)}$. But adding any such edge $e$ to the graph *decreases* the value of the partition function in (2.1) (the sum is over fewer independent sets), thereby increasing the likelihood of each of the samples.

The sample complexity and computational complexity of SIMPLEHC is as follows, with proof in the Supplement.

**Theorem 2.1.** *Consider the hard-core model* (2.1) *on a graph* $G = (V, E)$ *on* $|V| = p$ *nodes and with maximum degree* $d$. *The sample complexity of* SIMPLEHC *is*

$$n = O((2\lambda)^{2d-2} \log p), \tag{2.2}$$

*i.e. with this many samples the algorithm* SIMPLEHC *correctly reconstructs the graph with probability* $1 - o(1)$. *The computational complexity is*

$$O(np^2) = O((2\lambda)^{2d-2}p^2 \log p).$$ (2.3)

We next show that the sample complexity bound in Theorem 2.1 is basically tight:

**Theorem 2.2** (Sample complexity lower bound). *Consider the hard-core model* (2.1). *There is a family of graphs on $p$ nodes with maximum degree $d$ such that for the probability of successful reconstruction to approach one, the number of samples must scale as*

$$n = \Omega\left((2\lambda)^{2d} \log \frac{p}{d}\right).$$

**Lemma 2.3.** *Suppose edge $e = (i, j) \notin G$, and let $I$ be an independent set chosen according to the Gibbs distribution* (2.1). *Then* $\mathsf{P}(\{i, j\} \subseteq I) \geq (9 \cdot \max\{1, (2\lambda)^{2d-2}\})^{-1} \triangleq \gamma$.

The Supplementary Material contains proofs for Theorem 2.2 and Lemma 2.3.

## 3  Learning anti-ferromagnetic Ising models

In this section we consider the anti-ferromagnetic Ising model on a graph $G = (V, E)$. We parametrize the model in such a way that each configuration has probability

$$\mathsf{P}(\sigma) = \frac{1}{Z} \exp\{H(\sigma)\}, \quad \sigma \in \{0, 1\}^p,$$ (3.1)

where

$$H(\sigma) = -\beta \sum_{(i,j) \in E} \sigma_i \sigma_j + \sum_{i \in V} h_i \sigma_i.$$ (3.2)

Here $\beta > 0$ and $\{h_i\}_{i \in V}$ are real-valued parameters, and we assume that $|h_i| \leq h$ for all $i$. Working with configurations in $\{0, 1\}^p$ rather than the more typical $\{-1, +1\}^p$ amounts to a reparametrization (which is without loss of generality as shown for example in Appendix 1 of [25]). Setting $h_i = h = \ln \lambda$ for all $i$, we recover the hard-core model with fugacity $\lambda$ in the limit $\beta \to \infty$, so we think of (3.2) as a "soft" independent set model.

### 3.1  Strongly antiferromagnetic models

We start by considering the situation in which the repelling strength $\beta$ is sufficiently large that we can modify the approach used for the hard-core model. We require some notation to work with conditional probabilities: for each vertex $b \in V$, let

$$B_b = \{\sigma^{(i)} : \sigma_b^{(i)} = 1\},$$

and

$$\hat{\mathsf{P}}(\sigma_a = 1 | \sigma_b = 1) := \frac{1}{|B|}|\{i \in B : \sigma_a^{(i)} = 1\}|.$$

Of course, $\mathsf{E}(\hat{\mathsf{P}}(\sigma_a = 1 | \sigma_b = 1)) = \mathsf{P}(\sigma_a = 1 | \sigma_b = 1)$. The algorithm, described next, determines whether each edge $\{a, b\}$ is present based on comparing $\hat{\mathsf{P}}$ to a threshold.

---

**Algorithm 2** STRONGREPELLING

---

Input: $\beta$, $h$, $d$, and $n$ samples $\sigma^{(1)}, \ldots, \sigma^{(n)} \in \{0, 1\}^p$. Output: edge set $\hat{E}$.
1: Let $\delta = (1 + 2^d e^{h(d-1)})^{-2}$
2: FOR each possible edge $\{a, b\} \in \binom{V}{2}$:
3:   IF $\hat{P}(\sigma_a = 1 | \sigma_b = 1) \leq (1 + e^{\beta - h})^{-1} + \delta$ THEN add edge $(a, b)$ to $\hat{E}$
4: OUTPUT $\hat{E}$

---

Algorithm STRONGREPELLING obtains the following performance. The proof of Proposition 3.1 is similar to that of Theorem 2.1, replacing Lemma 2.3 by Lemma 3.2 below.

**Proposition 3.1.** *Consider the antiferromagnetic Ising model* (3.2) *on a graph* $G = (V, E)$ *on* $p$ *nodes and with maximum degree* $d$. *If*

$$\beta \geq d(h + \ln 2),$$

*then algorithm* STRONGREPELLING *has sample complexity*

$$n = O\left(2^{2d} e^{2h(d+1)} \log p\right),$$

*i.e. this many samples are sufficient to reconstruct the graph with probability* $1 - o(1)$. *The computational complexity of* STRONGREPELLING *is*

$$O(np^2) = O\left(2^{2d} e^{2h(d+1)} p^2 \log p\right).$$

When the interaction parameter $\beta \geq d(h + \ln 2)$ it is possible to identify edges using pairwise statistics. The next lemma, proved in the Supplement, shows the desired separation.

**Lemma 3.2.** *We have the following estimates:*

(i) *If* $(a, b) \notin E(G)$, *then* $\mathsf{P}(\sigma_a = 1 | \sigma_b = 1) \geq \frac{1}{1 + 2^{\deg(a)} e^{h(\deg(a) + 1)}}$.

(ii) *Conversely, if* $(a, b) \in E(G)$, *then* $\mathsf{P}(\sigma_a = 1 | \sigma_b = 1) \leq \frac{1}{1 + e^{\beta - h}}$.

(ii) *For any* $b \in V$, $\mathsf{P}(\sigma_b = 1) \geq \frac{1}{1 + 2^{\deg(b)} e^{h(\deg(b) + 1)}}$.

## 3.2  Weakly antiferromagnetic models

In this section we focus on learning weakly repelling models and show a trade-off between computational complexity and strength of the repulsion. Recall that for strongly repelling models our algorithm has run-time $O(p^2 \log p)$, the same as for the hard-core model (infinite repulsion).

For a subset of nodes $U \subseteq V$, let $G \backslash U$ denote the graph obtained from $G$ by removing nodes in $U$ (as well as any edges incident to nodes in $U$). The following corollary is immediate from Lemma 3.2.

**Corollary 3.3.** *We have the conditional probability estimates for deleting subsets of nodes:*

(i) *If* $(a, b) \notin E(G)$, *then for any subset of nodes* $U \subset V \setminus \{a, b\}$,

$$\mathsf{P}_{G \backslash U}(\sigma_a = 1 | \sigma_b = 1) \geq \frac{1}{1 + 2^{\deg_{G \backslash U}(a)} e^{h(\deg_{G \backslash U}(a) + 1)}}.$$

(ii) *Conversely, if* $(a, b) \in E(G)$, *then for any subset of nodes* $U \subseteq V \setminus \{a, b\}$

$$\mathsf{P}_{G \backslash U}(\sigma_a = 1 | \sigma_b = 1) \leq \frac{1}{1 + e^{\beta - h}}.$$

We can effectively remove nodes from the graph by conditioning: The family of models (3.2) has the property that conditioning on $\sigma_i = 0$ amounts to removing node $i$ from the graph.

**Fact 3.4** (Self-reducibility). *Let* $G = (V, E)$, *and consider the model* 3.2. *Then for any subset of nodes* $U \subseteq V$, *the probability law* $\mathsf{P}_G(\sigma \in \cdot | \sigma_U = \mathbf{0})$ *is equal to* $\mathsf{P}_{G \backslash U}(\sigma_{V \backslash U} \in \cdot)$.

The final ingredient is to show that we can condition by restricting attention to a subset of the observed data, $\sigma^{(1)}, \ldots, \sigma^{(n)}$, without throwing away too many samples.

**Lemma 3.5.** *Let* $U \subseteq V$ *be a subset of nodes and denote the subset of samples with variables* $\sigma_U$ *equal to zero by* $A_U = \{\sigma^{(i)} : \sigma_u^{(i)} = 0 \text{ for all } u \in U\}$. *Then with probability at least* $1 - \exp(n/2(1 + e^h)^{2|U|})$ *the number* $|A_U|$ *of such samples is at least* $\frac{n}{2} \cdot (1 + e^h)^{-|U|}$.

We now present the algorithm. Effectively, it reduces node degree by removing nodes (which can be done by conditioning on value zero), and then applies the strong repelling algorithm to the residual graph.

**Algorithm 3** WEAKREPELLING

Input: $\beta$, $h$, $d$, and $n$ samples $\sigma^{(1)}, \dots, \sigma^{(n)} \in \{0,1\}^p$. Output: edge set $\hat{E}$.
1: Let $\delta = (1 + 2^d e^{h(d-1)})^{-2}$
2: FOR each possible edge $(a,b) \in \binom{V}{2}$:
3:     FOR each $U \subseteq V \setminus \{a,b\}$ of size $|U| \leq \lceil d - \beta/(h + \ln 2) \rceil$
4:         Compute $\hat{P}_{G \setminus U}(\sigma_a = 1 | \sigma_b = 1)$
5:     IF $\min_{U:|U|=} \hat{P}_{G \setminus U}(\sigma_a = 1 | \sigma_b = 1) \leq (1 + e^{\beta - h}) + \delta$ THEN add edge $(a,b)$ to $\hat{E}$
6: OUTPUT $\hat{E}$

**Theorem 3.6.** *Let $\alpha$ be a nonnegative integer strictly smaller than $d$, and consider the antiferromagnetic Ising model 3.2 with*

$$\beta \geq (d - \alpha)(h + \ln 2)$$

*on a graph $G$. Algorithm WEAKREPELLING reconstructs the graph with probability $1 - o(1)$ as $p \to \infty$ using*

$$n = O\left((1 + e^h)^\alpha 2^{2d} e^{2h(d+1)} \log p\right)$$

*i.i.d. samples, with run-time*

$$O\left(np^{2+\alpha}\right) = \widetilde{O}_{h,d}(p^{2+\alpha}).$$

## 4   Statistical algorithms and proof of Theorem 1.1

We start by describing the statistical algorithm framework introduced by [1]. In this section it is convenient to work with variables taking values in $\{-1, +1\}$ rather than $\{0, 1\}$.

### 4.1   Background on statistical algorithms

Let $\mathcal{X} = \{-1, +1\}^p$ denote the space of configurations and let $\mathcal{D}$ be a set of distributions over $\mathcal{X}$. Let $\mathcal{F}$ be a set of solutions (in our case, graphs) and $\mathcal{Z} : \mathcal{D} \to 2^{\mathcal{F}}$ be a map taking each distribution $D \in \mathcal{D}$ to a subset of solutions $\mathcal{Z}(D) \subseteq \mathcal{F}$ that are defined to be valid solutions for $D$. In our setting, since each graphical model is identifiable, there is a single graph $\mathcal{Z}(D)$ corresponding to each distribution $D$. For $n > 0$, the *distributional search problem* $\mathcal{Z}$ over $\mathcal{D}$ and $\mathcal{F}$ using $n$ samples is to find a valid solution $f \in \mathcal{Z}(D)$ given access to $n$ random samples from an unknown $D \in \mathcal{D}$.

The class of algorithms we are interested in are called *unbiased statistical algorithms*, defined by access to an unbiased oracle. Other related classes of algorithms are defined in [1], and similar lower bounds can be derived for those as well.

**Definition 4.1** (Unbiased Oracle)**.** Let $D$ be the true distribution. The algorithm is given access to an oracle, which when given any function $h : \mathcal{X} \to \{0, 1\}$, takes an independent random sample $x$ from $D$ and returns $h(x)$.

These algorithms access the sampled data only through the oracle: unbiased statistical algorithms outsource the computation. Because the data is accessed through the oracle, it is possible to prove *unconditional* lower bounds using information-theoretic methods. As noted in the introduction, many algorithmic approaches can be implemented as statistical algorithms.

We now define a key quantity called average correlation. The *average correlation* of a subset of distributions $\mathcal{D}' \subseteq \mathcal{D}$ relative to a distribution $D$ is denoted $\rho(\mathcal{D}', D)$,

$$\rho(\mathcal{D}', D) := \frac{1}{|\mathcal{D}'|^2} \sum_{D_1, D_2 \in \mathcal{D}'} \left| \left\langle \frac{D_1}{D} - 1, \frac{D_2}{D} - 1 \right\rangle_D \right|, \tag{4.1}$$

where $\langle f, g \rangle_D := \mathsf{E}_{x \sim D}[f(x)g(x)]$ and the ratio $D_1/D$ represents the ratio of probability mass functions, so $(D_1/D)(x) = D_1(x)/D(x)$.

We quote the definition of statistical dimension with average correlation from [1], and then state a lower bound on the number of queries needed by any statistical algorithm.

**Definition 4.2** (Statistical dimension). Fix $\gamma > 0, \eta > 0$, and search problem $\mathcal{Z}$ over set of solutions $\mathcal{F}$ and class of distributions $\mathcal{D}$ over $X$. We consider pairs $(D, \mathcal{D}_D)$ consisting of a "reference distribution" $D$ over $\mathcal{X}$ and a finite set of distributions $\mathcal{D}_D \subseteq \mathcal{D}$ with the following property: for any solution $f \in \mathcal{F}$, the set $\mathcal{D}_f = \mathcal{D}_D \setminus \mathcal{Z}^{-1}(f)$ has size at least $(1 - \eta) \cdot |\mathcal{D}_D|$. Let $\ell(D, \mathcal{D}_D)$ be the largest integer $\ell$ so that for any subset $\mathcal{D}' \subseteq \mathcal{D}_f$ with $|\mathcal{D}'| \geq |\mathcal{D}_f|/\ell$, the average correlation is $|\rho(\mathcal{D}', D)| < \gamma$ (if there is no such $\ell$ one can take $\ell = 0$). The *statistical dimension* with average correlation $\gamma$ and solution set bound $\eta$ is defined to be the largest $\ell(D, \mathcal{D}_D)$ for valid pairs $(D, \mathcal{D}_D)$ as described, and is denoted by $\mathrm{SDA}(\mathcal{Z}, \gamma, \eta)$.

**Theorem 4.3** ([1]). *Let $\mathcal{X}$ be a domain and $\mathcal{Z}$ a search problem over a set of solutions $\mathcal{F}$ and a class of distributions $\mathcal{D}$ over $\mathcal{X}$. For $\gamma > 0$ and $\eta \in (0, 1)$, let $\ell = \mathrm{SDA}(\mathcal{Z}, \gamma, \eta)$. Any (possibly randomized) unbiased statistical algorithm that solves $\mathcal{Z}$ with probability $\delta$ requires at least $m$ calls to the Unbiased Oracle for*

$$m = \min \left\{ \frac{\ell(\delta - \eta)}{2(1 - \eta)}, \frac{(\delta - \eta)^2}{12\gamma} \right\} .$$

*In particular, if $\eta \leq 1/6$, then any algorithm with success probability at least $2/3$ requires at least $\min\{\ell/4, 1/48\gamma\}$ samples from the Unbiased Oracle.*

In order to show that a graphical model on $p$ nodes of maximum degree $d$ requires computation $p^{\Omega(d)}$ in this computational model, we therefore would like to show that $\mathrm{SDA}(\mathcal{Z}, \gamma, \eta) = p^{\Omega(d)}$ with $\gamma = p^{-\Omega(d)}$.

## 4.2 Soft parities

For any subset $S \subset [p]$ of cardinality $|S| = d$, let $\chi_S(x) = \prod_{i \in S} x_i$ be the parity of variables in $S$. Define a probability distribution by assigning mass to $x \in \{-1, +1\}^p$ according to

$$p_S(x) = \frac{1}{Z} \exp(c \cdot \chi_S(x)) . \tag{4.2}$$

Here $c$ is a constant, and the partition function is

$$Z = \sum_x \exp(c \cdot \chi_S(x)) = 2^{p-1}(e^c + e^{-c}) . \tag{4.3}$$

Our family of distributions $\mathcal{D}$ is given by these soft parities over subsets $S \subset [p]$, and $|\mathcal{D}| = \binom{p}{d}$. The following lemma, proved in the supplementary material, computes correlations between distributions.

**Lemma 4.4.** *Let $U$ denote the uniform distribution on $\{-1, +1\}^p$. For $S \neq T$, the correlation $\langle \frac{p_S}{U} - 1, \frac{p_T}{U} - 1 \rangle$ is exactly equal to zero for any value of $c$. If $S = T$, the correlation $\langle \frac{p_S}{U} - 1, \frac{p_S}{U} - 1 \rangle = 1 - \frac{4}{(e^c + e^{-c})^2} \leq 1$.*

**Lemma 4.5.** *For any set $\mathcal{D}' \subseteq \mathcal{D}$ of size at least $|\mathcal{D}|/p^{d/2}$, the average correlation satisfies $\rho(\mathcal{D}', U) \leq d^d p^{-d/2} .$*

*Proof.* By the preceding lemma, the only contributions to the sum (4.1) comes from choosing the same set $S$ in the sum, of which there are a fraction $1/|\mathcal{D}'|$. Each such correlation is at most one by Lemma 4.4, so $\rho \leq 1/|\mathcal{D}'| \leq p^{d/2}/|\mathcal{D}| = p^{d/2}/\binom{p}{d} \leq d^d/p^{d/2}$. Here we used the estimate $\binom{n}{k} \geq (\frac{n}{k})^k$. □

*Proof of Theorem 1.1.* Let $\eta = 1/6$ and $\gamma = d^d p^{-d/2}$, and consider the set of distributions $\mathcal{D}$ given by soft parities as defined above. With reference distribution $D = U$, the uniform distribution, Lemma 4.5 implies that $\mathrm{SDA}(\mathcal{Z}, \gamma, \eta)$ of the structure learning problem over distribution (4.2) is at least $\ell = p^{d/2}/d^d$. The result follows from Theorem 4.3. □

## Acknowledgments

This work was supported in part by NSF grants CMMI-1335155 and CNS-1161964, and by Army Research Office MURI Award W911NF-11-1-0036.

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
