[Supplementary Material]

## Supplementary material

**Proof of Theorem 2.1**

Algorithm SIMPLEHC correctly reconstructs the graph $G$ if for every edge $e = \{i, j\}$ not in $E(G)$, at least one observed independent set vector $\sigma^{(k)}$ contains both $i$ and $j$. Let $A^k_{ij} = \{\sigma^{(k)}_i = 0 \text{ or } \sigma^{(k)}_j = 0\}$ be the event that at least one of $i$ or $j$ is missing from $\sigma^{(k)}$, and let $A_{ij} = \cap^n_{k=1} A^k_{ij}$. We have by the union bound and independence of $A^k_{ij}$ for different $k$,

$$\mathsf{P}(\text{error}) \leq \mathsf{P}\left(\bigcup_{(i,j)\in E^c} A_{ij}\right) \leq \binom{p}{2}\mathsf{P}(\cap^n_{k=1}A^k_{ij}) = \binom{p}{2}\mathsf{P}(A^1_{ij})^n \leq \binom{p}{2}(1-\gamma)^n.$$

The last inequality is from Lemma 2.3, with the value of $\gamma$ the quantity in the statement of the Lemma. To make $\mathsf{P}(\text{error})$ approach zero at the rate $1/p$ it suffices to take $n = 3\gamma^{-1}\log p$. This proves the theorem. $\qquad\square$

**Proof of Theorem 2.2**

Consider the set of graphs $\mathcal{G}_m$ obtained by taking an arbitrary graph on $m$ nodes with maximum degree $d$, and to each vertex $v$ adding $d$ nodes $u_1, \ldots, u_d$ with edges $\{v, u_i\}$. The total number of nodes is $p = m(d+1)$. Thus we are working with the set of graphs consisting of $m = p/(d+1)$ stars of degree $d$, with all remaining edges going between centers of stars.

The goal is to determine the correct subset of the $\binom{m}{2}$ remaining edges. Fix a constant $c > 0$ and consider any graph $G \in \mathcal{G}_m$ missing at least $cm$ edges. Note that such graphs consist of almost all of $\mathcal{G}_m$ (a proportion $1 - o(1)$).

We bound the number of samples required by the maximum-likelihood (ML) rule (equivalent to algorithm SIMPLEHC) to reconstruct $G$. As observed in Section 2, the ML graph contains the edge $e = \{i, j\}$ between star centers $i$ and $j$ if and only if none of the sets $\sigma^{(1)}, \sigma^{(2)}, \ldots, \sigma^{(n)}$ contains both $i$ and $j$. Thus, in order for ML to give the correct graph, for each missing edge $e = \{i, j\}$ it is necessary to observe a "witness" $\sigma^{(k)}$ with $\sigma^{(k)}_i = \sigma^{(k)}_j = 1$.

We proceed by upper bounding the probability of observing a witness for each of the $cm$ missing edges. Each star center $i$ is included in a particular random independent set $\sigma^{(k)}$ with probability at most

$$q = \frac{1}{2 \cdot (2\lambda)^d},$$

hence $\sigma^{(k)}$ is a witness for missing edge $\{i, j\}$ with probability at most $q^2$. Hence the expected number of missing edges which within $n$ samples have no witness is at least $cm(1 - q^2)^n$, and a second moment argument shows that one must take

$$n \geq (1 + o(1))\frac{\log m}{-\log(1 - q^2)} = \Omega\left((2\lambda)^{2d}\log m\right),$$

where we used the fact that $-\log(1 - q^2) = q^2 + o(q^4)$ and $q^{-1} = (2\lambda)^d$. $\qquad\square$

**Proof of Lemma 2.3**

We can decompose the partition function as

$$Z_G = \sum_I \lambda^{|I|} = \sum_{I\in S_{\varnothing,\varnothing}} \lambda^{|I|} + \sum_{I\in S_{\varnothing,j}} \lambda^{|I|} + \sum_{I\in S_{i,\varnothing}} \lambda^{|I|} + \sum_{I\in S_{i,j}} \lambda^{|I|}$$

$$:= Z_{\varnothing,\varnothing} + Z_{\varnothing,j} + Z_{i,\varnothing} + Z_{i,j}, \tag{4.4}$$

where $S_{ij} = \{I : i, j \in I\}$, $S_{i,\varnothing} = \{I : i \in I, j \notin I\}$, etc. Now, $Z_G$ and $Z_{G+e}$ are the same except $Z_{G+e}$ does not have the last term $Z_{i,j}$. We bound the last term by first noting that

$$|S_{i,j}| \cdot 2^d \geq |S_{\varnothing,j}|. \tag{4.5}$$

This is because for each independent set $I$ with $i \in I$, there are at most $2^d$ distinct independent sets $I'$ with $i \notin I'$ with some subset of (at most $d$) neighbors of $i$ included. One way of observing this is defining the map $f : S_{\varnothing,j} \to S_{i,j}$ by $I \mapsto \{i\} \cup I \setminus \mathcal{N}(i)$. The map $f$ takes at most $2^d$ sets $I' \in S_{\varnothing,j}$ to each $I \in S_{i,j}$, which implies (4.5).

Now, each such set $I'$ mapping to $I$ has weight at most a factor $\lambda^{d-1}$ larger than $I$, so

$$2^d \lambda^{d-1} Z_{i,j} \geq Z_{\varnothing,j}. \tag{4.6}$$

Similar reasoning gives

$$2^d \lambda^{d-1} Z_{i,j} \geq Z_{i,\varnothing}, \quad \text{and} \quad 2^{2d} \lambda^{2d-2} Z_{i,j} \geq Z_{\varnothing,\varnothing}. \tag{4.7}$$

Using these estimates, we obtain

$$\mathsf{P}(\{i,j\} \subseteq I) = \frac{\sum_{I:\{i,j\}\subseteq I} \lambda^{|I|}}{\sum_I \lambda^{|I|}} = \frac{Z_{i,j}}{Z} \geq \frac{1}{1 + 4 \cdot (2\lambda)^{d-1} + 4 \cdot (2\lambda)^{2d-2}},$$

proving the lemma. $\qquad \square$

### Proof of Lemma 3.2

We start by defining restricted partition function summations: Let

$$S_{ab} = \{\sigma \in \{0,1\}^p : \sigma_a = \sigma_b = 1\},$$
$$S_{a\varnothing} = \{\sigma \in \{0,1\}^p : \sigma_a = 1, \sigma_b = 0\},$$

and analogously for $S_{\varnothing b}$ and $S_{\varnothing\varnothing}$. We then define $Z_{ab} = \sum_{\sigma \in S_{ab}} \exp(H(\sigma))$ and again analogously for $Z_{a\varnothing}, Z_{\varnothing b}, Z_{\varnothing\varnothing}$.

We first prove case (i) of the lemma, in which we assume that $(a,b) \notin E(G)$ and lower bound the probability

$$\mathsf{P}(\sigma_a = 1 | \sigma_b = 1) = \frac{Z_{ab}}{Z_{ab} + Z_{\varnothing b}}.$$

To this end, consider the map $f : S_{\varnothing b} \to S_{ab}$ defined by taking a configuration $\sigma$, setting $\sigma_i = 0$ for neighbors $i \in N(a)$, and then setting $\sigma_a = 1$. Since the assumption $(a,b) \notin E(G)$ implies that $\sigma_a = \sigma_b = 1$ is a valid assignment to these variables, the definition of $f$ implies in particular that $(f(\sigma))_b = 1$ if $\sigma_b = 1$, so indeed $f(\sigma) \in S_{ab}$ for $\sigma \in S_{\varnothing b}$.

Now, at most $2^{\deg(a)}$ sets are mapped by $f$ to any one set (since the neighbors of $a$ can be in any configuration), and for any $\sigma \in S_{\varnothing b}$, $\exp(H(f(\sigma)) \geq \exp(H(\sigma) - h(\deg(a) + 1))$. This shows that $2^{\deg(a)} \exp[h(\deg(a) + 1)] Z_{ab} \geq Z_{\varnothing b}$, and proves part (i) of the lemma.

We now turn to case (ii), assuming that $(a,b) \in E(G)$. Consider the map $g : S_{ab} \to S_{\varnothing b}$ taking $\sigma \in S_{ab}$ and setting $\sigma_a = 0$ (removing node $a$ from the independent set). The map $g$ is one-to-one, and $H$ increases by $\beta$ due to resolving the conflict on edge $(a,b)$, but decreases by $h_a \leq h$ due to omitting node $a$: $\exp(H(g(\sigma))) \geq \exp(H(\sigma) + \beta - h)$. This shows that $Z_{ab} \geq e^{-\beta+h} Z_{\varnothing b}$, and completes the proof. $\qquad \square$

### Proof of Lemma 3.5

We start by computing the probability that a particular sample $\sigma^{(i)}$ is in $A_U$, or equivalently that $\sigma_U = \mathbf{0}$. Let $W \subseteq V$ be any subset of nodes, and denote by $x_W$ an assignment to the corresponding variables. Due to the antiferromagnetic nature of the interaction, the distribution (3.2) satisfies the monotonicity property $\mathsf{P}(\sigma_a = 1 | \sigma_W = x_W) \leq \mathsf{P}(\sigma_a = 1 | \sigma_W = x_W, \sigma_b = 0)$ for any neighbor $b \in N(a) \setminus W$. This monotonicity together with Bayes' rule gives

$$\mathsf{P}(\sigma_U = \mathbf{0}) = \prod_{i=1}^{|U|} \mathsf{P}(\sigma_{u_i} = 0 | \sigma_{u_1} = \cdots = \sigma_{u_{i-1}} = 0) \geq \prod_{i=1}^{|U|} \mathsf{P}(\sigma_{u_i} = 0 | \sigma_{N(u_i)} = \mathbf{0})$$

$$= \prod_{i=1}^{|U|} [1 + e^{h_i}]^{-1}.$$

Denoting the last displayed quantity by $q$, we see that the number of samples obtained, $|A_U|$, stochastically dominates a $\text{Binom}(n, q)$ random variable. Hoeffding's inequality proves the lemma. □

**Proof of Lemma 4.4**

Calculating correlation relative to the uniform distribution $U$ (see Equation (4.1)), we have for $S \neq T$ with $|S \cap T| = \lambda$

$$\left\langle \frac{p_s}{U} - 1, \frac{p_T}{U} - 1 \right\rangle_U = \sum_{x \in \{-1,+1\}^p} 2^{-p}(2^p p_S(x) - 1)(2^p p_T(x) - 1)$$

$$= \sum_{x \in \{-1,+1\}^p} 2^p p_S(x) p_T(x) - 1. \tag{4.8}$$

Now

$$\sum_{x \in \{-1,+1\}^p} 2^p p_S(x) p_T(x) = \frac{2^p}{Z^2} \sum_x \exp(c \cdot (\chi_S(x) + \chi_T(x)))$$

$$= \frac{2^p \cdot 2^{p-2N+\lambda}}{Z^2} \sum_{x_{S \cap T}} \sum_{x_{S \triangle T}} \exp(c \cdot (\chi_S(x) + \chi_T(x)))$$

$$\stackrel{(a)}{=} \frac{2^p \cdot 2^{p-2N+\lambda}}{Z^2} \sum_{x_{S \cap T}} 2^{2N-2\lambda} \cdot \frac{1}{4} \cdot \left(e^{2c} + e^{-2c} + 2\right)$$

$$= \frac{2^{2p-2}}{Z^2}(e^c + e^{-c})^2 \stackrel{(b)}{=} 1.$$

Step (a) follows from the fact that for any fixed $x_{S \cap T}$, half the assignments to $x_{S \setminus T}$ result in $\chi_S = 1$ and half $\chi_S = -1$, and similarly for $x_{T \setminus S}$; step (b) is from the formula (4.3) for $Z$.

For the case $S = T$, we have

$$\sum_{x \in \{-1,+1\}^p} 2^p p_S(x) p_T(x) = \frac{2^p}{Z^2} \sum_x \exp(c \cdot (\chi_S(x) + \chi_T(x)))$$

$$= \frac{2^p \cdot 2^{p-1}}{Z^2}(e^{2c} + e^{-2c})$$

$$= \frac{2^{2p-2}}{Z^2} 2(e^c + e^{-c})^2 - \frac{4}{(e^c + e^{-c})^2} = 2 - \frac{4}{(e^c + e^{-c})^2} .$$

Plugging this into (4.8) completes the proof. □