[Reviews · NeurIPS 2014]

Submitted by Assigned_Reviewer_13

This is a theory heavy paper regarding the structure learning of antiferromagnetic Ising models. There are two main results in this paper. First, the authors, for the class of statistical algorithms introduced by Feldman et al, provided a computational lower bound for learning general graphical models on p nodes with maximum degree d. Second, the authors showed that a broad class of repelling models on general graphs can be learned using simple algorithms, even without the correlation decay property.
The results, to the best of my knowledge, are new.
The paper is well structured and written.
The paper would be even better if the authors can highlight the impact of the results (i.e. why these results are important).

Summary: A good paper. The paper would be even better if the authors can highlight the impact of the results.

Submitted by Assigned_Reviewer_44

This paper presents some theoretical results for the complexity of learning pairwise graphical models. Arguably, the most important result is a lower bound of the complexity of the general case. I believe the initial part of the paper can be improved. For instance, a "graphical model" is never defined and the discussion of sect.1.1 and 1.2 are confusing because of that. While I understand what the authors want to say, they should formally define the problem of interest, including the graphical model and the objective function that they want to optimise as being the problem of "learning the structure of general graphical models". For instance (and if I understood correctly), the work considers graphical models with pairwise relations only, which could be emphasised in the initial description of the problem (currently missing, even though the paper later clarify most of my doubts, reading its initial part for the first time was not much informative). The goal is explained as "recovering a graph on p nodes", but this is too informal. The problem that is being solved only becomes clear in sect.2, and even there it is not really formally defined. Because of the presentation, theorem 1.1 is not so appealing in the first time one reads it. In a different note, the initial complexity discussion ignores amount of data to learn, although this is clarified later.

All in all, the paper seems sound and with an interesting theoretical study. However, I admit not to be an expert on this exact topic, and it is not easy to me to assess well the relevance and novelty. For example, the main result (of theorem 1.1) does not seem surprising to me, but if it had not been proved before, its proof is of clear relevance.

Minor issues:

- "Thus algorithms... does" -> do

- "turns out reconstruct" ?

- Algorithm 1: S should start empty, algorithm 2 and 3, hat{E}, etc. Better at least to make clear that sets are assumed to start empty.

- "Thus the ML estimate contains a subset of those edges e which have not..." Could you explain this with slightly more detail?

Summary: Theoretical paper about learning Ising models. Main result is a lower bound for learning such graphs that almost matches a previously known upper bound. Results seem sound and interesting, although I can't judge its relevance well.

Submitted by Assigned_Reviewer_46

This paper describes new results for learning undirected graphical models: a lower bound on learning general models using a parity example, and an algorithm for learning repelling models whose complexity depends on the strengh of repulsion.

The results are significant and original, and the paper is clearly written. I did not read the supplementary material, and I'm assuming everything is correct. I was not familiar with the hard-core model, and I struggled a bit connecting the problem to the notion of "independent sets"; I think this could have benefited from a more gentle explanation, but the authors are struggling with space.
Summary: Important new results clearly presented.
Author Feedback
Author rebuttal: We thank the reviewers for their comments. The main concerns of the reviewers pertained to the relevance and possible impact of the work.

The problem of structure learning of graphical models is practically important and has received a massive amount of attention in the machine learning community over the last decade. A central challenge is to find computationally efficient structure learning algorithms. In this paper we consider the problem of learning graphs of maximum degree d on p nodes. The exhaustive search algorithm of Bresler et al. (APPROX 2008) can learn in time roughly p^d, but this is not practically feasible for large values of d. Therefore, a relevant notion of computational tractability is requiring time of the order f(d) p^c, where f(d) is some constant depending on d, and the exponent c is independent of degree.

Most papers on structure learning propose an algorithm and attempt to find conditions under which the algorithm learns correctly. It is difficult to compare the performance of different algorithms, and there is no baseline to determine if some models are somehow inherently difficult to learn. In this paper we advocate a different approach to make progress on the structure learning problem: we seek to characterize for which model parameters learning is tractable or intractable.

Our first result is a lower bound of the form p^(d/2) on the computation time of learning general graphical models on graphs with maximum degree d. This implies that the structure learning problem is not tractable in general. The lower bound is for the restricted class of algorithms known as “statistical algorithms”, introduced recently by Feldman et al. (STOC 2013) in order to show lower bounds for the planted clique problem; this class of algorithms encompasses nearly every known algorithmic approach, so the lower bound carries some weight.

In light of our general lower bound, in order to do structure learning efficiently it seems necessary to focus on restricted classes of models. There are two types of model assumptions: structural assumptions on the graph (e.g., tree or tree-like) and assumptions on the parameters. Aside from structural assumptions, to the best of our knowledge all previous structure learning algorithms require the model parameters to be chosen so that the model satisfies the so-called correlation decay property, which roughly means that far apart variables are nearly independent (see the paper by Bento and Montanari, NIPS 2009). We identify a new class of models which violate the correlation decay property in a very strong way, so-called repelling models, for which structure learning is nevertheless an algorithmically tractable problem. We further introduce a simple algorithm that is able to interpolate between p^2 log p computation, achievable for models with strong repulsion, and p^d, achievable in general, depending on the strength of the repulsion in the model.

We believe that the approach taken in this paper has the potential to create a major impact in graphical model learning. By focusing on specific model parameter regimes, new algorithms can be developed with an eye towards the challenging, and as yet unsolved, model classes. At the same time, we show how the recently proposed lower-bound methodology of Feldman can be applied to give computational lower bounds for learning graphical models.